# IRM—when it works and when it doesn't:
# A test case of natural language inference

**Yana Dranker[1]**
yanadr@campus.technion.ac.il

**He He[2]**
hhe@nyu.edu

**Yonatan Belinkov[1]\***
belinkov@technion.ac.il

[1] Technion – Israel Institute of Technology          [2] New York University

## Abstract

Invariant Risk Minimization (IRM) is a recently proposed framework for out-of-distribution (o.o.d) generalization. Most of the studies on IRM so far have focused on theoretical results, toy problems, and simple models. In this work, we investigate the applicability of IRM to bias mitigation—a special case of o.o.d generalization—in increasingly naturalistic settings and deep models. Using natural language inference (NLI) as a test case, we start with a setting where both the dataset and the bias are synthetic, continue with a natural dataset and synthetic bias, and end with a fully realistic setting with natural datasets and bias. Our results show that in naturalistic settings, learning complex features in place of the bias proves to be difficult, leading to a rather small improvement over empirical risk minimization. Moreover, we find that in addition to being sensitive to random seeds, the performance of IRM also depends on several critical factors, notably dataset size, bias prevalence, and bias strength, thus limiting IRM's advantage in practical scenarios. Our results highlight key challenges in applying IRM to real-world scenarios, calling for a more naturalistic characterization of the problem setup for o.o.d generalization.

## 1   Introduction

Deep learning models show strong performance when tested on data from the same distribution they were trained on, matching or even surpassing humans (Zhang et al., 2017; Brinker et al., 2019). However, this performance often stems from relying on spurious correlations rather than human-like reasoning, causing these models to "break" in real world scenarios (Geirhos et al., 2018; McCoy et al., 2019). A recent method called Invariant Risk Minimization (IRM; Arjovsky et al. 2020) aims to learn causal features whose correlation with the label is invariant across different distributions, thus leading to better out-of-distribution (o.o.d) generalization. IRM splits the training data into different subsets, or *environments*, across which varying spurious correlations cause distribution shifts. The goal is to learn a representation that yields the same optimal classifier for all environments.

Although several studies (Choe et al., 2020; Ahuja et al., 2020a; Kamath et al., 2021) have investigated IRM and its variants, the empirical results so far have mainly focused on synthetic settings and simpler models (e.g., shallow multilayer-perceptrons, bag-of-words models). This motivates us to investigate IRM in natural settings with complex models. We take natural language inference (NLI) as a test case, where the model needs to predict if a hypothesis sentence is entailed by the premise sentence. While theoretically requiring deep language understanding, many NLI datasets contain biases, or spurious correlations between superficial features and labels (Gururangan et al., 2018; Poliak et al., 2018; Tsuchiya, 2018), such as word overlap correlating with the entailment label. Such heuristics allow models to perform superficially well on the benchmark, but fail catastrophically when these heuristics

---

\*Supported by the Viterbi Fellowship in the Center for Computer Engineering at the Technion.

35th Conference on Neural Information Processing Systems (NeurIPS 2021).

no longer hold (Naik et al., 2018; Glockner et al., 2018; McCoy et al., 2019). Common debiasing methods typically rely on explicit modeling of the biases (Belinkov et al., 2019a,b; Karimi Mahabadi et al., 2020; He et al., 2019). In contrast, IRM offers interesting possibilities. It is model agnostic, not built for a specific bias and, given appropriate environments, generalizes to distributions where the spurious correlation varies.

In this work, we design a series of debiasing experiments, intended to bridge the gap between fully synthetic and naturalistic scenarios. Table 1 shows examples of each setting. We begin with a toy experiment where both the dataset and the bias are synthetic. Next, we add a level of complexity by injecting synthetic bias to a real NLI dataset and use state-of-the-art pre-trained models. Last, we investigate natural NLI datasets with known dataset biases. For the last setting, splitting environments using categorical biased features as done in previous work is not applicable. Therefore, for real-world scenarios where the bias is known but might be a high dimensional feature, we develop a simple way to split data into different environments, described in "Environment generation" in Section 3.2. In each of these settings, we train models with empirical risk minimization (ERM) and IRM and evaluate them on o.o.d test sets. Our experiments yield the following results:

- In the toy setting, performance follows the theory: ERM performs well on the training set but fails completely on the o.o.d test set, while IRM ignores the bias and thus performs slightly worse on the training set, but perfectly on the o.o.d test set.

- On natural NLI datasets and deep models—with either synthetic bias or natural bias—IRM outperforms ERM when evaluated on o.o.d test sets.

- However, in these more naturalistic settings, IRM is not able to completely discard the bias, while ERM does not rely solely on the bias. Thus, in practice the advantage of IRM is small.

- When used with state-of-the-art models, IRM shows a large variance in performance across random seeds.

To better characterize when IRM works better or worse than ERM, we highlight three criteria that govern its success: the prevalence of the bias in the training set (how many examples are biased), the strength of the bias (how strong is the correlation between a label and a biased feature), and the training data size. While bias strength and data size were not empirically investigated in real-world scenarios, bias prevalence was completely overlooked in previous work. We find that when all three criteria are met (prevalent and strong bias, large training set), IRM tends to perform better than ERM. When bias prevalence, strength, or training data size are limited, IRM is less stable and ERM performs better.

To conclude, our investigation shows important challenges in assessing the performance of IRM in plausible real-world scenarios, pointing to the need to employ a more naturalistic approach to characterizing the settings in which IRM can perform well.

## 2   Related Work

**IRM**   IRM was motivated by Peters et al. (2016), who connect invariance with causality by establishing a link between the cause of a target variable and the invariant predictor. As a proof-of-concept, Arjovsky et al. experimented only with very simple settings, namely MNIST with varying background colors. Several studies further investigated IRM or proposed variants of IRM. Rosenfeld et al. (2020) show that in the non-linear setting, there exists a nearly optimal classifier which uses non-invariant features, thus the classifier found by IRM may not generalize to o.o.d. data. However, their result assumes infinite samples and we instead focus on an empirical study in the finite sample case. Kamath et al. (2021) point out the discrepancies between the original IRM formulation and various instantiations of it. Since in non-toy examples such comparison is non-trivial, we focus on IRMv1, the practical formulation suggested in the original paper. Both Ahuja et al. (2020a) and Krueger et al. (2021) propose new objectives and show superior stability, and improvement over IRM when adding covariate shift, respectively, on colored MNIST. Gulrajani and Lopez-Paz (2020) devise a test suite to fairly compare algorithms for domain generalization, and show that with careful tuning and model selection, ERM outperforms all other algorithms. Choe et al. (2020) show that IRM outperforms ERM in a sentiment analysis task with synthetic bias, but restrict themselves to a bag-of-words model.

Table 1: Example NLI samples for all proposed settings. In the training set, biased features $B$ are correlated with the label $y$ (same color). In the test set, the correlation does not hold (different colors).

| Data | Bias | Training example | Test example |
|---|---|---|---|
| | | **Toy Experiment** | |
| Synthetic | Synthetic | $y$: non-entailment
$B$: has "c"
$P$: a    $H$: bc | $y$: entailment
$B$: has "c"
$P$: a    $H$: ac |
| | | **Synthetic Bias Experiment** | |
| Natural | Synthetic | $y$: contradiction
$B$: has "<c>"
$P$: A man resting on a street.
$H$: <c> A man jogging on the street. | $y$: entailment
$B$: has "<c>"
$P$: A car sinking in water.
$H$: <c> A car is flooding. |
| | | **Natural Bias Experiment (overlap bias)** | |
| Natural | Natural | $y$: entailment
$B$: has word overlap
$P$: It is most natural.
$H$: It is natural. | $y$: non-entailment
$B$: has word overlap
$P$: Don't be a cynic.
$H$: Be a cynic. |
| | | **Natural Bias Experiment (hypothesis bias)** | |
| Natural | Natural | $y$: contradiction
$B$: has negation
$P$: Three bikers stop in town.
$H$: The bikers didn't stop in the town. | $y$: entailment
$B$: has negation
$P$: A barefoot woman reads a book [...]
$H$: A woman isn't wearing any shoes. |

While different approaches were suggested, most are evaluated in simple settings. Although Gulrajani and Lopez-Paz targeted more naturalistic scenarios, they focused only on image classification. In addition to focusing on a language task, we show performance gains of IRM over ERM in naturalistic settings with complex models, and perform an analysis of the factors that affect IRM's performance.

**Bias in NLI**   NLI is a widely studied task in natural language processing, concerned with identifying the relation between a premise and a hypothesis. Recent studies (Gururangan et al., 2018; Poliak et al., 2018; McCoy et al., 2019) reveal biases in NLI datasets such as SNLI (Bowman et al., 2015) and MNLI (Williams et al., 2018). Two widely observed biases are hypothesis bias and overlap bias. Overlap bias is characterized by high word overlap between the premise and the hypothesis, shown to be correlated with entailment. Hypothesis bias refers to patterns in the hypothesis correlated with a specific label, allowing for correct prediction without considering the premise; a common example is negation words correlated with the contradiction label (Table 1). Consequently, several challenge sets (Naik et al., 2018; Glockner et al., 2018; McCoy et al., 2019) demonstrated performance degradation when the bias no longer holds. We evaluate the ability of IRM to mitigate reliance on bias and thus improve performance in such cases.

## 3   Method

### 3.1   Background on IRM

Our goal is to investigate the ability of IRM (Arjovsky et al., 2020) to debias NLI models and improve their o.o.d generalization. IRM attempts to base its prediction on features whose correlation with the target variable is invariant rather than environment-specific. This is translated to finding a data representation such that the optimal classifier on top of it is the same across environments:

$$\min_{\substack{\Phi:\mathcal{X}\to\mathcal{H} \\ w:\mathcal{H}\to\mathcal{Y}}} \sum_{e\in\mathcal{E}_{tr}} R^e(w\circ\Phi) \quad \text{s.t.} \quad w\in\arg\min_{w':\mathcal{H}\to\mathcal{Y}} R^e(w'\circ\Phi) \quad \forall e\in\mathcal{E}_{tr}. \tag{1}$$

where $\mathcal{E}_{tr}$ is the set of the training environments, $R^e$ is the risk of environment $e$, $w$ is the classifier in the hypothesis space $\mathcal{H}$, and $\Phi$ is the representation function. This optimization problem is relaxed into a regularized objective function called IRMv1, and to which we simply refer as IRM:

$$\min_{\Phi:\mathcal{X}\to\mathcal{Y}} \sum_{e\in\mathcal{E}_{tr}} R^e(\Phi) + \lambda \cdot \|\nabla_{w|_{w=1.0}} R^e(w\cdot\Phi)\|^2, \tag{2}$$

where the hypothesis class is now limited to linear classifiers, and our goal is to find a representation function $\Phi$ for which the classifier is optimal in all environments. The first term, referred to as the ERM term, promotes low average error over all environments, while the second term, referred to as the IRM term, promotes invariance across environments. The constraint from Eq. 1 is relaxed into a Lagrangian form that penalizes the gradient norm of each environment. The regularizer weight $\lambda$ controls the trade-off between empirical risk and invariance.

### 3.2   Problem Setup and Experimental Details

**Terminology and notation**   Let $\mathcal{D}_{tr} = \{(x_i, y_i)\}_{i=1}^N$ denote a training dataset of $N$ premise–hypothesis samples $x = (P, H)$ with labels $y \in \{\text{entailment}, \text{neutral}, \text{contradiction}\}$, and an associated dataset bias. A bias is a correlation between some feature of a sample $x = (P, H)$ and a label $y$, that is not guaranteed to hold outside the dataset. We call such a feature a biased feature and denote it as $B(x)$. For instance, $B(x)$ might be word overlap in the case of overlap bias or negation words in the case of hypothesis bias. We call a sample $x_i$ with label $y_i$ bias aligned if $p(y = y_i \mid B(x_i))$ in the training set is high. Similarly, it is bias misaligned if $p(y = y_i \mid B(x_i))$ in the training set is low. Finally, unbiased samples, as opposed to biased samples, are such for which the feature $B(x)$ is not correlated with any of the labels. Given an environment $\mathcal{E}_{tr_e}$ (a subset of $\mathcal{D}_{tr}$), we refer to the conditional probability of a label given the biased feature, $p(y = y' \mid B(x))$, as the bias *strength* in the environment and denote it as $p_e$. A related but different quantity is the ratio of biased samples out of all samples in the environment, which we call the bias *prevalence* and denote as $\alpha_e$.

**Environment generation**   The goal of IRM is to find a representation function $\Phi$ of the premise–hypothesis pair, for which the classifier is optimal in all environments. In our case, we generate environments by varying the bias strength, i.e., changing the conditional probability of the label given the biased feature. In the toy and the synthetic bias experiments the bias is synthetically injected into the samples, so generating environments is straightforward. For a bias feature $B(x)$ and a label $y'$, to create an environment $e$ with $p(y = y' \mid B(x)) = p_e$, we associate the feature with the label with probability $p_e$. This fully controllable setting no longer holds when we proceed to the natural bias case, where the bias cannot be decoupled from the samples.

In order to generate environments that represent a varying bias strength for some $B(x)$ in the natural bias experiments, we first need to quantify $p(y = y' \mid B(x))$. We train a model $f_\theta$ with $B(x)$ as its only input, referred to as a *biased model*. For instance, this might be a model with access only to the hypothesis, in the case of hypothesis bias. The biased model outputs a probability distribution over the labels $f_\theta(B(x)) \in \mathcal{R}^{|\mathcal{Y}|}$, termed a *score vector*. The model prediction $\hat{y}_i$ (the label receiving the highest probability) and the score vector are used to assign the sample to one of three subsets—$\mathcal{D}_{un,B}$ (unbiased), $\mathcal{D}_{align,B}$ (bias aligned) or $\mathcal{D}_{misalign,B}$ (bias misaligned)—as follows.

First, we categorize samples to unbiased and biased. A sample is considered *unbiased* if the total variation distance of $f_\theta(B)$ from the uniform distribution is lower than some threshold $t_1$. Otherwise, it is considered biased. Next, the subsets $\mathcal{D}_{align,B}$ and $\mathcal{D}_{misalign,B}$ are constructed from the biased samples such that $\mathcal{D}_{align,B} = \{x_i \mid \hat{y}_i = y_i\}$ and $\mathcal{D}_{misalign,B} = \{x_i \mid \hat{y}_i \neq y_i\}$. Lastly, to ensure that the biased feature is highly indicative of the label, we filter the subsets $\mathcal{D}_{align,B}, \mathcal{D}_{misalign,B}$ by eliminating examples where the top two predictions are very close. Denote by $\tilde{y}_i$ the second highly predicted label for the $i$-th example. The resulting sets are $\mathcal{D}_{align,B} = \{x_i \mid \hat{y}_i = y_i \wedge \hat{y}_i - \tilde{y}_i > t_2\}$, $\mathcal{D}_{misalign,B} = \{x_i \mid \hat{y}_i \neq y_i \wedge \hat{y}_i - \tilde{y}_i > t_2\}$ for some threshold $t_2$.

To generate an environment $e$ with bias strength $p_e$ and bias prevalence $\alpha_e$, we sample $(m, n, k)$ examples from the subsets $(\mathcal{D}_{align,B}, \mathcal{D}_{misalign,B}, \mathcal{D}_{un,B})$ such that $p_e = \frac{m}{m+n}$ and $\alpha_e = \frac{m+n}{m+n+k}$.[2] This process will be used to generate environments with pre-defined bias strength in Section 4 and

---

[2]We first choose $m$ and $n$ as large as possible such that required strength holds and resulting environments are of equal size. Then we add the $k$ samples as required by the prevalence.

pre-defined bias strength and prevalence in the natural bias experiments in Section 5. Models are trained on environments $\mathcal{E}_{tr}$ generated from $\mathcal{D}_{tr}$ and evaluated on test environments generated from splits of an appropriate test set. First, $\mathcal{E}_{id,B}$ is a test environment with $\alpha_e = 1.0$ and $p_e$ similar to what was set in the training environments. Thus, it represents a similar distribution to $\mathcal{E}_{tr}$ w.r.t the bias $B$, and considered in-distribution. Second, $\mathcal{E}_{ood,B}$ is a test environment with $\alpha_e = 1.0$ and $p_e$ reversed to what was set in the training environments (i.e., if $p_e$ was high in $\mathcal{E}_{tr}$, it will be low in $\mathcal{E}_{ood,B}$). Appropriately, $\mathcal{E}_{ood,B}$ represents a (often significantly) different distribution w.r.t the bias, and considered o.o.d. Third, $\mathcal{E}_{un,B}$ is a test set with unbiased samples, i.e., $\alpha_e = 0.0$. For brevity, we will omit $B$ when it is clear from context.

**Training details**    All the experiments feature two training environments, both of the same size, on which the models are trained. In the case of ERM, where we do not have an additional regularization term ($\lambda = 0$), this is equivalent to training on a mixture of data from the two environments. Table 6 in Appendix A describes the environment specification for each experiment in details. According to the practical instantiation of IRM (Eq. 2), the classifier in all the experiments is a "dummy" classifier ($w = 1.0$ or a vector of ones for multi-class classification), and the representation function $\Phi$ is the model used in the experiment. For more experimental details refer to Appendix A. Our code and data can be found at `https://github.com/technion-cs-nlp/irm-for-nli`.

# 4   Results

## 4.1   Toy Experiment

### 4.1.1   Setup

We design a XOR example as a simplified NLI task, and use it to simulate hypothesis bias. To construct our synthetic dataset, we adapt the example from Belinkov et al. (2019b), where the premise and hypothesis are each one character long taken from the set $\{a, b\}$. The premise entails the hypothesis ($y = 1$) if their first character is the same. These samples are randomly partitioned to two disjoint subsets, from which we build environments $\mathcal{E}_{tr_e}$ ($e \in \{1, 2\}$) as follows: first add noise to the label (by flipping it with probability $\eta_e$), then append 'c' and 'd' to the hypothesis of non-entailment and entailment examples with high probability $p_e$, respectively, such that the model can take the shortcut to predict by the appended character. By setting $p_1 = 0.8$, $p_2 = 0.9$, we construct two training environments with a strong yet varying correlation between the label and the bias. This correlation is flipped at test time, by setting $p_e = 0.0$ and $\eta_e = 0.0$. Note that noise is needed when training to make the correlation of the label with the biased feature stronger than its correlation with the causal feature (i.e. equality of the first character) ($1 - \eta_e < p_e$), such that ERM will rely on the biased feature. To summarize, we train both ERM and IRM on two training environments $\mathcal{E}_{tr_1}$, $\mathcal{E}_{tr_2}$ with $\alpha_1 = \alpha_2 = 1.0$, $p_1 = 0.8$, and $p_2 = 0.9$. The models are then evaluated on a test environment $\mathcal{E}_{ood}$ with $\alpha_1 = \alpha_2 = 1.0$ and $p_e = 0.0$. We use the same model described in Belinkov et al. (2019b)—a multi-layered perceptron (MLP) on character embeddings—trained with cross-entropy loss. For a detailed description of the model and hyper-parameters, see Appendix B.

### 4.1.2   Results

Theoretically, relying on the hypothesis bias (the appended character) to predict entailment should result in approximately $\frac{p_1 + p_2}{2} = 85\%$ accuracy on the combined training set (since the classes are balanced), and $0\%$ accuracy on the test set. Relying on the unbiased feature, on the other hand, should result in approximately $1 - \eta = 75\%$ accuracy on the training set, and $100\%$ accuracy on the test set. As Table 2 shows, our results confirm these expectations. ERM relies on the appended character to predict the label, thus failing completely on the test set. IRM manages to identify the environment-specific correlation and relies on the causal feature, achieving $100\%$ test accuracy. An analysis of the training dynamics of ERM and IRM (Appendix B) shows that IRM starts shifting the weight from the biased feature to the causal feature when the constrained phase (high $\lambda$) begins.

Table 2: Accuracy in the toy experiment.

| | Train | Test |
|---|---|---|
| ERM | $85.304 \pm 0.4$ | $0.0 \pm 0.0$ |
| IRM | $75.384 \pm 0.69$ | $100.0 \pm 0.0$ |

Table 3: Accuracy in the synthetic bias experiment.

| | $p_e = 0.8$ | $p_e = 0.33$ | $p_e = 0.0$ |
|---|---|---|---|
| REF | $89.97 \pm 0.34$ | $89.33 \pm 0.28$ | $88.93 \pm 0.55$ |
| ERM | $93.49 \pm 0.28$ | $85.16 \pm 0.9$ | $79.16 \pm 1.48$ |
| IRM | $92.32 \pm 0.3$ | $87.22 \pm 0.45$ | $83.5 \pm 0.71$ |

## 4.2 Synthetic Bias Experiment

### 4.2.1 Setup

In this section we add the complexity of a real world dataset (SNLI) and use a state-of-the-art model. We are interested in how well IRM works with a high capacity model, previously shown to rely on bias (McCoy et al., 2019; Poliak et al., 2018). This experiment is inspired by a similar experiment suggested in He et al. (2019) and designed to simulate hypothesis bias. We use three bias tokens (`<c>`, `<e>`, and `<n>`), correlated with the three labels in SNLI (contradiction, entailment, and neutral, respectively). [3] We prepend the bias token corresponding to the gold label with probability $p_e$, and randomly sample one of the other two tokens otherwise. We use pre-trained BERT (base-uncased) as our model, with most of the hyper-parameters as recommended in the original work (Devlin et al., 2019). For further training details, see Appendix C.1.

We train our models on two training environments $\{\mathcal{E}_{tr_e}\}$ ($e \in \{1, 2\}$) with $\alpha_1 = \alpha_2 = 1.0$, $p_1 = 0.7$ and $p_2 = 0.9$. We additionally include a REF model trained on the original SNLI training set with no bias injected (i.e., $\mathcal{E}_{tr_e}$ with $\alpha_e = 0.0$), thus considered an unbiased model (w.r.t the synthetic bias). The training environments are of equal size and comprise together the entire SNLI training/validation set. The models are evaluated on three environments generated from the SNLI test set—$\mathcal{E}_{id}$ with $p_e = \frac{p_1+p_2}{2} = 0.8$, $\mathcal{E}_{ood}$ with $p_e = 0.33$ and $\mathcal{E}_{ood}$ with $p_e = 0.0$, where the latter comes down to a bias misaligned set. All test environments are set with $\alpha = 1.0$.

### 4.2.2 Results

The results in Table 3 show that IRM outperforms ERM on o.o.d test environments and degrades on the in-distribution test environment, indicating that it successfully reduces reliance on the biased feature. However, as the test environment moves further away from the training environments (i.e., more bias misaligned samples or decreasing $p_e$), both ERM and IRM display lower accuracy, revealing that IRM is not able to completely ignore the bias when the test environments are sufficiently different. As expected, the unbiased model (REF) performs similarly across all environments. This performance can serve as both a lower bound for the in-distribution degradation and an upper bound for the o.o.d improvement, and indeed we can see that both ERM and IRM performance are higher than REF on $p_e = 0.8$ and lower on $p_e = 0.33, 0.0$.

Relying solely on the biased feature should have resulted in performance close to chance (33%) on the $p_e = 0.33$ environment, but both ERM and IRM perform significantly better. Comparing the accuracy of an unbiased model (REF model) on test environment $\mathcal{E}_{un,B}$ ($\sim 90\%$) with the expected accuracy of a model relying strictly on the biased feature to predict ($\sim \frac{p_1+p_2}{2} = 80\%$), suggests that unbiased features are more correlated with the label than the biased feature, discouraging ERM from heavily relying on the biased feature. Combined with IRM's inability to completely discard the bias, this leads to IRM showing rather small performance gains over ERM. Finally, in Appendix C.3 we also show the results when injecting the entire test set with the same bias token. This experiment reveals that ERM's performance is more strongly degraded by the neutral bias token, suggesting that biases in the data may be treated differently. In contrast, IRM's performance is similar given any of the three bias tokens.

---

[3] We have experimented with several sets of bias tokens and witnessed similar behavior. Further details in Appendix C.2.

Table 4: Accuracy on SNLI test environments for hypothesis bias.

|  | Unbiased | Bias aligned | Bias misaligned |
|---|---|---|---|
| ERM | 84.46 (±0.64) | 97.4 (±0.26) | 62.63 (±1.19) |
| IRM | 82.64 (±1.33) | 91.4 (±2.92) | 65.12 (±2.28) |

Table 5: F1 macro on MNLI dev mismatched test environments for overlap bias.

|  | Unbiased | Bias aligned | Bias misaligned |
|---|---|---|---|
| ERM | 85.23 (±0.69) | 96.67 (±0.28) | 62.66 (±1.95) |
| IRM | 83.75 (±0.46) | 95.44 (±1.1) | 64.12 (±3.86) |

## 4.3 Natural Bias Experiment

### 4.3.1 Setup

In this experiment our goal is to create environments that vary by known dataset bias. We target two widely observed biases—hypothesis bias and overlap bias (Section 2)—on two large NLI datasets, SNLI and MNLI. Following previous work (He et al., 2019; Karimi Mahabadi et al., 2020), we use a BERT model with hypothesis as its only input as a biased model for hypothesis bias and a shallow MLP on top of manually-designed features as a biased model for overlap bias (Appendix D.2). Since the overlap feature is not discriminative between neutral and contradiction classes (McCoy et al., 2019; Karimi Mahabadi et al., 2020), we combine neutral and contradiction into a single non-entailment class. As this results in an unbalanced dataset, all models are trained with loss weighted according to class size and we report macro F1 scores instead of accuracy. For further training details, see Appendix D.1.

For both hypothesis and overlap bias we train our models on two training environments $\{\mathcal{E}_{tr_e}\}$ ($e \in \{1, 2\}$) with $p_1 = 0.7$ and $p_2 = 0.9$. Similarly to the approach in previous work (Arjovsky et al., 2020; Krueger et al., 2021; Choe et al., 2020), we only control the bias strength of the environments. The bias prevalence is not fixed, and the resulting average prevalence of the two environments is similar to the bias prevalence found in the original dataset without any interventions. For hypothesis bias we have $\frac{\alpha_1 + \alpha_2}{2} = 0.82$ and for overlap bias $\frac{\alpha_1 + \alpha_2}{2} = 0.52$. We evaluate our models on in distribution and o.o.d test environments — $\mathcal{E}_{id}$ with $\alpha_e = p_e = 1.0$ (bias aligned subset) and $\mathcal{E}_{ood}$ with $\alpha_e = 1.0$ and $p_e = 0.0$ (bias misaligned subset). We also report results on $\mathcal{E}_{un}$ with $\alpha_e = 0.0$, i.e., the unbiased subset. Size of training/validation data is $\sim$305k/ $\sim$5k and $\sim$245k/$\sim$6k for hypothesis and overlap bias, respectively.

### 4.3.2 Results

**Hypothesis bias** We report results for hypothesis bias on SNLI test environments in Table 4. The performance gap between the environments is significant, indicating that models strongly rely on biased features. Similarly to the observation in the synthetic bias case, IRM improves performance on the bias misaligned subset and degrades on the bias aligned subset compared to ERM, proving to be more robust. Although IRM displays some degradation on the unbiased subset, it is significantly smaller than the one on the bias aligned subset. We also note that performance varies more on the bias misaligned subset (larger std from different random seeds), which is consistent with results on o.o.d evaluation (McCoy et al., 2020).

The relatively poor performance (of both IRM and ERM) on the bias misaligned subset may be explained by an inherent difficulty of the samples in this subset. Indeed, we found that the inter-annotator agreement in this subset is much lower compared to the other subsets (Appendix D.3). This suggests that the causal features are noisier in the bias misaligned subset and that the upper bound on performance is lower than the two other sets.

**Overlap bias** Table 5 shows the results for overlap bias on MNLI dev mismatched environments.[4] Performance discrepancies are as expected, with highest performance on bias aligned subset and

---

[4]Since the MNLI test set labels are hidden, and are needed for splitting according to bias, we use the provided dev matched set as a validation set, and test our models on the dev mismatched set.

lowest on bias misaligned subset. IRM decreases this performance gap between the subsets: It outperforms ERM on the bias misaligned subset, and slightly degrades on the other two. As before, performance varies more on the bias misaligned subset, especially for IRM.

To further understand IRM's improvement on the bias misaligned subset we analyzed the performance per class in Appendix D.4. We found that IRM mitigates the bias for the non-entailment class (i.e., improves on no-overlap entailment examples) suggesting that IRM's improvement on the bias misaligned subset comes from mitigating the non-overlap bias; surprisingly, it slightly amplifies bias for the entailment class (i.e., degrades on high overlap non-entailment examples). This suggests that different biases in the data might be unevenly mitigated by IRM.

**Evaluation on o.o.d benchmarks**  We evaluate our models trained in the overlap bias setting and the hypothesis bias setting on HANS (McCoy et al., 2019) and RevisedPremises (RP; Kaushik et al. (2019)) respectively, and report results in Appendix D.5. On RP test set, even though our models are trained on smaller data size, we achieve comparable results to those reported by Kaushik et al. (2019). On HANS, our results are not comparable to those presented in McCoy et al. (2019) for several reasons—the size of the training data is smaller in our case (due to constraints on environment generation), the bias is stronger by construction, and the labels are collapsed to 2 classes before training. On both benchmarks IRM is not able to outperform ERM, and performs equivalently (on RP) or worse (on HANS). We hypothesize that this is due to differences in biases between the o.o.d subsets we construct and the challenge benchmarks. This shows that while IRM might improve on biases targeted by the environments it was trained on, it is not able to extrapolate to different types of biases.

## 5   Analysis

In this section, we empirically analyze three factors that influence IRM's performance: bias prevalence, strength, and training data size. In each case, we vary one factor while keeping the other two fixed. Where possible, we use the default values from the main experiments in the previous section. However, to achieve the required bias prevalence and strength in the natural bias setting, we had to reduce the size of training and validation data significantly. As we shall see, the size of the training data is in itself a factor affecting IRM's performance, so this might influence the stability of the results. In addition to the previously observed sensitivity of IRM to random seeds (Ahuja et al., 2020a), which causes high variation (as shown in the previous section), we also witness another type of failure case with the restricted environments generated for the natural bias analysis. Some of the runs for IRM seem to converge to a null representation, displaying near-chance performance on both training and validation. The ERM term (Eq. 2) should have prevented this scenario, however tuning $\lambda$ to mitigate this behavior is an open issue. We therefore report results for all 5 random seeds when necessary.

### 5.1   Bias Prevalence

To analyze the effect of bias prevalence on IRM, we vary the fraction of biased samples in the training set, while fixing the bias strength ($p_1 = 0.7$, $p_2 = 0.9$) and the size of training/validation data ($\sim$134k/$\sim$2.6k for hypothesis bias, $\sim$87k/$\sim$2.3k for overlap bias, and $\sim$549k/$\sim$9.8k, i.e., the entire SNLI train/val set, for synthetic bias). We evaluate on $\mathcal{E}_{ood}$ with $\alpha = 1.0$ and $p_e = 0.0$ (i.e., the bias misaligned set) for all three settings. Figure 1 shows that as the bias prevalence increases, IRM's performance on o.o.d data improves. In the synthetic bias case, IRM surpasses ERM at high bias prevalence. In the natural bias cases, IRM has several failed runs (very low accuracy), probably due to the reduced training set size; however, those that succeeded, surpass ERM. A possible explanation is that as more samples are either bias aligned or bias misaligned (i.e. contain a biased feature), we get more signal in the IRM penalty term indicating the instability of the biased feature across environments, making it easier for IRM to detect and discard it.

### 5.2   Bias Strength

Similarly to the previous experiment, when analyzing the effect of bias strength we keep the bias prevalence fixed, ($\alpha_1 = \alpha_2 = 1.0$ for synthetic bias, $\alpha_1 = \alpha_2 = 0.82$ for hypothesis bias, and $\alpha_1 = \alpha_2 = 0.52$ for overlap bias). Size of training/validation data is $\sim$152k/$\sim$2.3k for hypothesis bias, $\sim$145k/$\sim$3.8k for overlap bias, and $\sim$549k/$\sim$9.8k (the entire SNLI train/val set) for synthetic

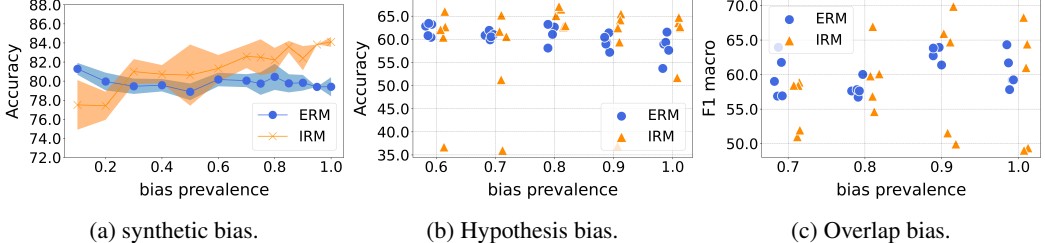

(a) synthetic bias.  (b) Hypothesis bias.  (c) Overlap bias.

Figure 1: Performance of ERM and IRM on $\mathcal{E}_{ood}$ with $p_e = 0.0$, $\alpha_e = 1.0$ (bias misaligned subset) when varying bias prevalence (holding size of data and bias strength fixed). Test subsets are generated from SNLI test for synthetic and hypothesis bias, and from MNLI dev mismatched for overlap bias.

bias. As before, we evaluate on $\mathcal{E}_{ood}$ with $\alpha_e = 1.0$ and $p_e = 0.0$ for synthetic bias, hypothesis bias, and overlap bias. Figure 2a shows results for the synthetic bias setting, indicating that as bias strength increases ERM degrades while IRM (roughly) maintains its performance. Figures 2b, 2c show results for both natural biases. As bias becomes stronger, ERM degrades on o.o.d, while most of the IRM runs significantly outperform ERM. Still, some of the IRM runs crash and perform poorly. We notice that while IRM seems to improve with increased prevalence, it degrades as bias strength increases, but just not as much as ERM. We hypothesize that while IRM is more robust to bias than ERM, it is not able to completely discard it, even in synthetic settings (see Section 4.2.2) let alone natural, more complex settings. Therefore, as bias is stronger it is picked up more by both methods and only partially discarded by IRM.

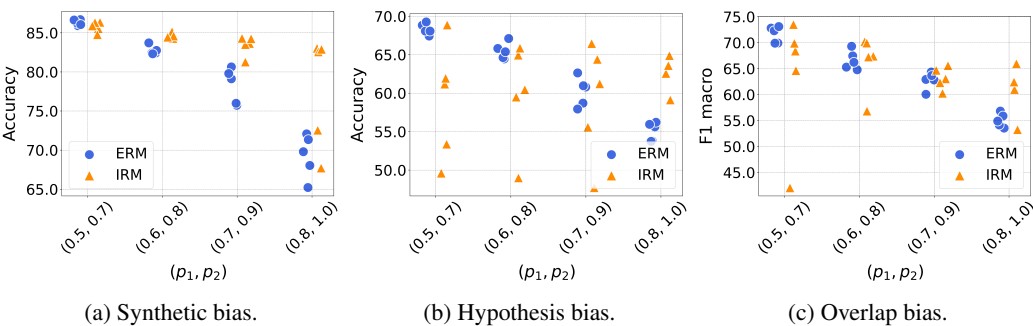

(a) Synthetic bias.  (b) Hypothesis bias.  (c) Overlap bias.

Figure 2: Performance of ERM and IRM on $\mathcal{E}_{ood}$ with $p_e = 0.0$, $\alpha_e = 1.0$ (bias misaligned subset) when varying bias strengths (holding size of data and bias prevalence fixed). Test subsets are generated from SNLI test for synthetic and hypothesis bias, and from MNLI dev mismatched for overlap bias.

## 5.3 Data Size

To isolate the effect of data size on performance, we keep the bias prevalence and strength fixed and equal to the experiments in Section 4.3 ($\alpha_1 = \alpha_2 = 1.0, 0.82, 0.52$ for synthetic bias, hypothesis bias, and overlap bias, respectively, $p_1 = 0.7$, $p_2 = 0.9$ for all settings), and only change the size of the environments. We evaluate on $\mathcal{E}_{ood}$ with $\alpha_e = 1.0$ and $p_e = 0.0$ for all three settings. For synthetic bias, Figure 3a shows that IRM performs on par or better than ERM on o.o.d data for all sizes, and loses stability as data size decreases. The gap between ERM's and IRM's performance seems to decrease as data size increases, possibly due to the simplicity of the synthetic bias, and the high correlation between true signal and label, discussed in Section 4.2.

For both natural biases, IRM is outperformed by ERM in small data regimes (Figures 3b and 3c). For hypothesis bias, IRM outperforms ERM on o.o.d data on several settings, while for overlap bias only in the largest regime is IRM able to outperform ERM. In Appendix D.2 we show that the biased model used for overlap bias generates less confident predictions, compared to that of the hypothesis bias. This might suggest that the overlap bias is less discriminative and negatively affects the quality of environments. This is also reflected in the lower prevalence in the overlap bias setting (0.52) compared to the hypothesis bias setting (0.82). Finally, the natural bias results fall in line with the

observation by Ahuja et al. (2020b) that, for linear and polynomial models, IRM, contrary to ERM, approaches the desired o.o.d solution in the finite sample regime as data size increases.

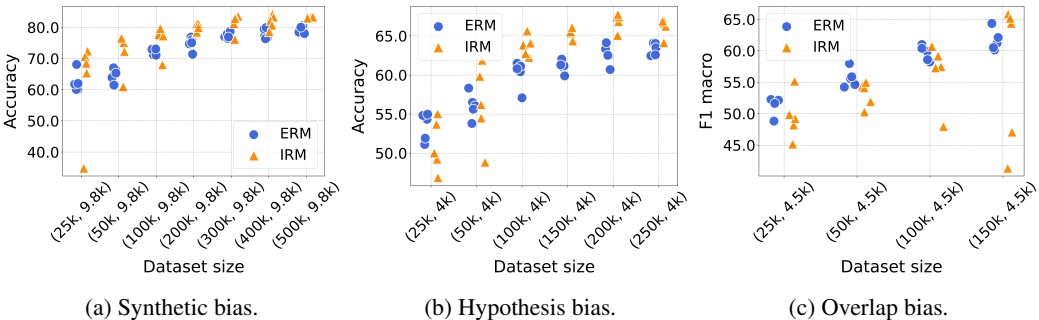

(a) Synthetic bias.        (b) Hypothesis bias.        (c) Overlap bias.

Figure 3: Performance of ERM and IRM on $\mathcal{E}_{ood}$ test subset when varying size of data (while holding bias prevalence and strength fixed). The X axis labels show (training data size, validation data size).

## 6    Conclusion

In this work, we presented an empirical study of IRM across increasingly naturalistic settings. We observed that in all three proposed settings IRM successfully outperforms ERM on o.o.d data. We then pointed out bias prevalence, strength, and data size as factors affecting IRM. We showed that IRM loses its advantage over ERM when decreasing any of these factors. Aside from IRM's known sensitivity to random seeds, we also observe a different failure case of IRM in which the model "collapses" to a null representation. As also mentioned in Arjovsky et al. (2020), such a degenerate solution should theoretically be discarded by the ERM term in the objective function. However, practically it is not yet clear how to properly change the regularization weight during training to avoid this. This failure case can be detected and filtered without o.o.d data, by only looking at performance on training and validation environments. However, for the sake of completeness we do not filter these results and report them in the paper. To conclude, IRM and other recent methods taking a causal approach to o.o.d generalization offer promising possibilities. Our results highlight the need for a more naturalistic characterization of the settings under which these methods are expected to work and an extensive empirical experimentation with complex datasets and state-of-the-art models.

## Broader impact and ethical considerations

Our work investigates IRM in a debiasing framework with naturalistic scenarios and characterizes its strengths and weaknesses. We hope that our results will encourage further research, both theoretical and empirical, to characterize and improve IRM's behavior in real-world settings. Such a line of work could contribute to an increased robustness of deep learning models in the presence of biases. While we focused on dataset biases in this study, the approach could also apply to mitigating social biases like gender or racial bias. There are, however, potential harms. One might assume that a debiased model may be used as is in a deployed system, without further safety measures. In practice, debiasing on academic datasets might not lead to safe-to-use models in real applications and could give a false impression of safety. Such a scenario somewhat corresponds to unconscious bias, which has been shown to exist in healthcare, criminal justice, and education. Compared with explicit bias, where measures promoting equality can be taken, unconscious bias goes undetected and therefore untreated. For example, Hall et al. (2015) found implicit bias in health care providers against people of colour, which negatively affected treatment decisions, treatment adherence, and patient health outcomes. We believe that carefully analyzing when IRM works and when it fails is important for guiding further research on debiasing natural language processing systems and machine learning systems more broadly.

## Acknowledgements

We would like to thank the anonymous reviewers for their helpful comments. This research was supported by the ISRAEL SCIENCE FOUNDATION (grant No. 448/20). HH is partially supported by Samsung Advanced Institute of Technology (Next Generation Deep Learning: From Pattern Recognition to AI).

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
