## A  Experimental Details

IRM is trained in two phases: a warm up phase, in which the regularizer weight ($\lambda$) is set to a small value to reach a desired area in the parameter space, and a constrained phase, in which the regularizer weight is set to a very large value to enforce the invariance constraint. All results are averaged over 5 runs with different random seeds, and mean and standard deviation are reported. Following previous recommendations (Gulrajani and Lopez-Paz, 2020; Teney et al., 2020) we do not use o.o.d as validation and we clearly specify our approach to model selection. In the synthetic and natural bias experiments, we construct in-distribution validation environments from the validation set (dev set for SNLI, dev matched for MNLI), using the specification as the training environments to ensure we do not leak o.o.d information into the validation process. Early stopping is applied in both these settings, stopping the training after a patience period if the loss on the validation environments does not decrease.

All models were optimised using AdamW (Loshchilov and Hutter, 2018) with $(\beta_1, \beta_2) = (0.9, 0.999)$ and weight decay 0.01. We make use of the `transformers` library (Wolf et al., 2020), and the pre-trained models supplied therein. All experiments were run on a GeForce RTX 2080 Ti GPU.

Table 6: Environment specification for each of the experiments. In the natural bias main experiments (Section 4.3), neither data size nor bias prevalence were fixed. The main experiments refer to the experiments reported in Section 4, while the bias strength, prevalence and data size refer to experiments performed in Section 5.

| experiment | $(p_1, p_2)$ | $(\alpha_1, \alpha_2)$ | train data size |
|---|---|---|---|
| Toy Experiment | | | |
| main | $(0.8, 0.9)$ | $(1.0, 1.0)$ | 20000 |
| Synthetic Bias Experiment | | | |
| main | $(0.7, 0.9)$ | $(1.0, 1.0)$ | 549k |
| bias strength | $\{[(x, x + 0.2)\|(x \in [0.5, 0.8]]\}$ | $(1.0, 1.0)$ | 549k |
| bias prevalence | $(0.7, 0.9)$ | $\{(x, x)\|x \in [0.1, 1.0]\}$ | 549k |
| data size | $(0.7, 0.9)$ | $(1.0, 1.0)$ | $[25k, 500k]$ |
| Natural Bias Experiment (overlap bias) | | | |
| main | $(0.7, 0.9)$ | $\frac{\alpha_1+\alpha_2}{2} = \frac{0.42+0.62}{2} = 0.52$ | 245k |
| bias strength | $\{[(x, x + 0.2)\|(x \in [0.5, 0.8]]\}$ | $(0.52, 0.52)$ | 145k |
| bias prevalence | $(0.7, 0.9)$ | $\{(x, x)\|x \in [0.7, 1.0]\}$ | 87k |
| data size | $(0.7, 0.9)$ | $(0.52, 0.52)$ | $[25k, 150k]$ |
| Natural Bias Experiment (hypothesis bias) | | | |
| main | $(0.7, 0.9)$ | $\frac{\alpha_1+\alpha_2}{2} = \frac{0.68+0.96}{2} = 0.82$ | 305k |
| bias strength | $\{[(x, x + 0.2)\|(x \in [0.5, 0.8]]\}$ | $(0.82, 0.82)$ | 152k |
| bias prevalence | $(0.7, 0.9)$ | $\{(x, x)\|x \in [0.6, 1.0]\}$ | 134k |
| data size | $(0.7, 0.9)$ | $(0.82, 0.82)$ | $[25k, 250k]$ |

# B  Toy Experiment

## B.1  Training details

The model used in the toy experiment follows the one used in Belinkov et al. (2019b). Both premise and hypothesis are represented as the sum of their character embeddings. These representations are concatenated and passed to a one-hidden-layer feed-forward network for binary classification. Both the embedding dimension and the hidden dimension of the hidden layer are set to 10. We train the model with a batch size of 1000 and evaluate every 5 batches. Both ERM and IRM are trained for a total of 120 steps, where the IRM training is divided to a warm up phase with 20 steps and a constrained phase with 100 steps.

## B.2  Prediction dynamics

We show the prediction dynamics during ERM and IRM training in Figures 4a and 4b, respectively. The samples in each of the plots have the same ground truth label but are appended each with a different synthetic token (i.e., 'c' or 'd'), revealing the difference in prediction depending on the added token. We can see that under the IRM regime (Figure 4b), two samples with the same true signal but different synthetic token get increasingly similar predictions (solid and dashed lines of the same color getting closer after the constrained phase begins). This is opposed to ERM, which makes a clear distinction based on the type of added token (solid and dashed lines stay separate).

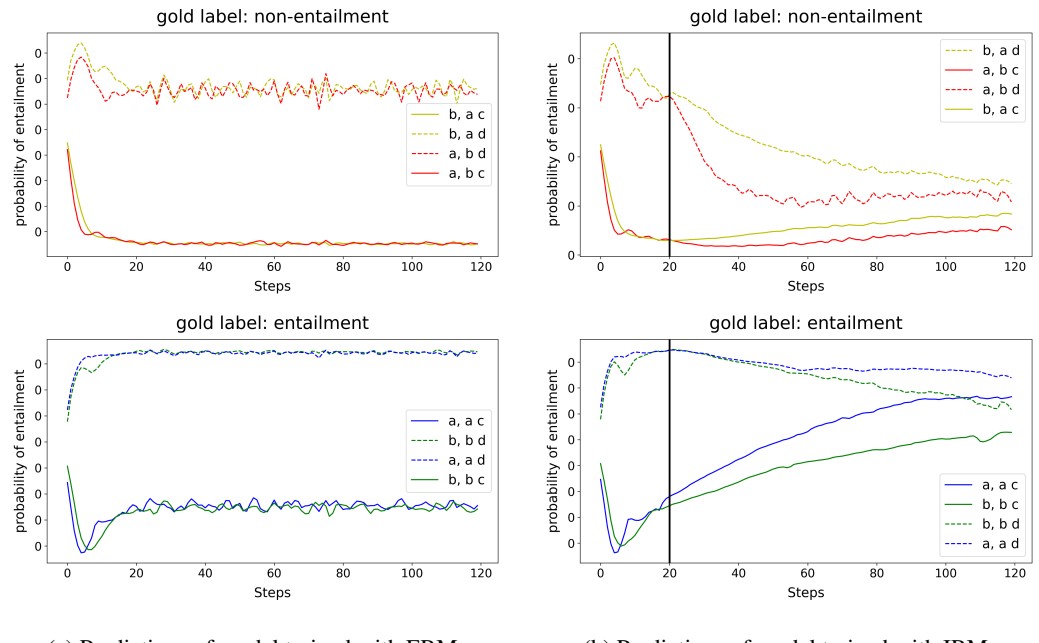

(a) Predictions of model trained with ERM.

(b) Predictions of model trained with IRM.

## C Synthetic bias experiment

### C.1 Training details

We specify training details for each of the models in the synthetic bias setting—REF, ERM, and IRM. We used batch size of 512 and learning rate of $5e^{-5}$ for all models. REF and ERM were trained for 4 epochs, with the regularizer weight set to $\lambda = 0.0$. IRM was trained in two phases (as explained in Section 3.2)—1 epoch for the warm up phase with $\lambda = 1.0$, and 4 epochs for the constrained phase with $\lambda = 1e^4$. BERT hyper-parameters not explicitly specified, like dropout, were unchanged from their default values.

### C.2 Effect of different bias tokens

We experimented with 3 additional sets of bias tokens. The bias tokens were chosen such that they appear in BERT's vocabulary but are absent from train, dev and test sets of the SNLI dataset. This is in order to disentangle learning of the biased correlation from learning other associations during fine-tuning. In all three cases we have witnessed results similar to those described in Section 4.2.2. Results for each of the sets is described in Table 7.

Table 7: Accuracy in the synthetic bias experiment for different sets of bias tokens.

| Bias tokens | | $p_e = 0.8$ | $p_e = 0.33$ | $p_e = 0.0$ |
|---|---|---|---|---|
| (walters, obligatory, rebecca) | ERM | $93.11 \pm 0.29$ | $84.99 \pm 0.43$ | $78.93 \pm 0.95$ |
| | IRM | $91.76 \pm 0.86$ | $86.69 \pm 0.93$ | $82.9 \pm 1.81$ |
| (contractual, viable, displaced) | ERM | $93.49 \pm 0.22$ | $85.14 \pm 0.57$ | $78.87 \pm 0.94$ |
| | IRM | $91.83 \pm 1.12$ | $87.0 \pm 0.42$ | $83.73 \pm 1.08$ |
| (millennium, scholarly, binary) | ERM | $93.45 \pm 0.32$ | $84.56 \pm 0.72$ | $78.26 \pm 1.03$ |
| | IRM | $91.32 \pm 0.46$ | $87.2 \pm 0.32$ | $84.44 \pm 0.13$ |

## C.3 Results per synthetic token

To understand the behavior of the models for each of the biases, we evaluate them on the test set injected entirely with the same synthetic token. Table 8 shows that injecting the token associated with the neutral label generates the largest degradation in performance for ERM, suggesting that the biased features are treated differently. We speculate that during training, the nature of neutral examples makes it harder for the model to classify them, leading it to rely more on the neutral bias feature. Thus, the model is affected more by the neutral bias than the other two biases.

Table 8: Accuracy on SNLI test set where all the samples are prepended with same synthetic token.

|  | contradiction bias | entailment bias | neutral bias |
|---|---|---|---|
| REF | $88.81 \pm 0.87$ | $89.48 \pm 0.35$ | $89.71 \pm 0.33$ |
| ERM | $85.37 \pm 0.96$ | $86.01 \pm 0.79$ | $84.33 \pm 1.13$ |
| IRM | $86.59 \pm 1.05$ | $87.22 \pm 0.46$ | $87.8 \pm 0.58$ |

## D  Natural bias experiment

### D.1  Training details

In table 9, we specify training details for each of the models in the natural bias setting. The table lists ERM, IRM, and the biased models that were used to generate scores for environment creation. ERM and IRM share the same training hyper-parameters for both hypothesis and overlap bias and are therefore mentioned in general. BERT hyper-parameters not explicitly specified are unchanged from their default values. Batch size was chosen to be as big as possible and required using gradient checkpointing.

Table 9: Training hyper-parameters for models in the natural bias setting.

|  | warm up epochs | epochs | warm up reg | reg | lr | batch size |
|---|---|---|---|---|---|---|
| ERM | 0 | 4 | 0.0 | 0.0 | $5e^{-5}$ | 512 |
| IRM | 1 | 4 | 1.0 | $1e^4$ | $5e^{-5}$ | 512 |
| hypothesis-biased model | 0 | 4 | 0.0 | 0.0 | $5e^{-5}$ | 512 |
| overlap-biased model | 0 | 25 | 0.0 | 0.0 | $1e^{-3}$ | 512 |

### D.2  Biased models and environment generation

The biased model used to generate the score vectors for hypothesis bias is BERT with only the hypothesis as input. For overlap bias, we used a shallow 3-layer MLP on top of manually designed features. We use the syntactic heuristics defined in McCoy et al. (2019) as features: lexical overlap (the ratio of overlapping words between premise and hypothesis and whether the premise contains all words used in the hypothesis), subsequence (is the hypothesis a contiguous subsequence of the premise), and constituent (is the hypothesis sub-tree of the premise parse tree). Similarly to Karimi Mahabadi et al. (2020), we also add similarity features between the premise and hypothesis representations (as generated by a pre-trained BERT model): the min, max, and mean of their dot product. To obtain score vectors from these models on the training set, we used k-fold cross validation ($k = 4$), each time training on $k - 1$ folds and scoring the left-out $k^{th}$ fold. The score vectors for development and test sets are mean over the score vectors given by the $k$ different models. The

hypothesis bias model achieved $70\%$ accuracy on the test set, while the overlap bias model achieved only $48\%$ accuracy.

In Figure 5 we present histograms of the score vectors to get a sense of the model's confidence. The histograms display the probability the biased model assigned to the ground truth label of the samples, binned into 10 bins of equal width. This is done for every label and each of the dataset subsets (train, validation and test). The hypothesis bias model displays many correct and confident predictions, specifically for the contradiction and neutral labels. The overlap bias model is not as confident and does not perform as well. We notice that for both the hypothesis and overlap bias models, the histograms across train, validation, and test subsets are similar.

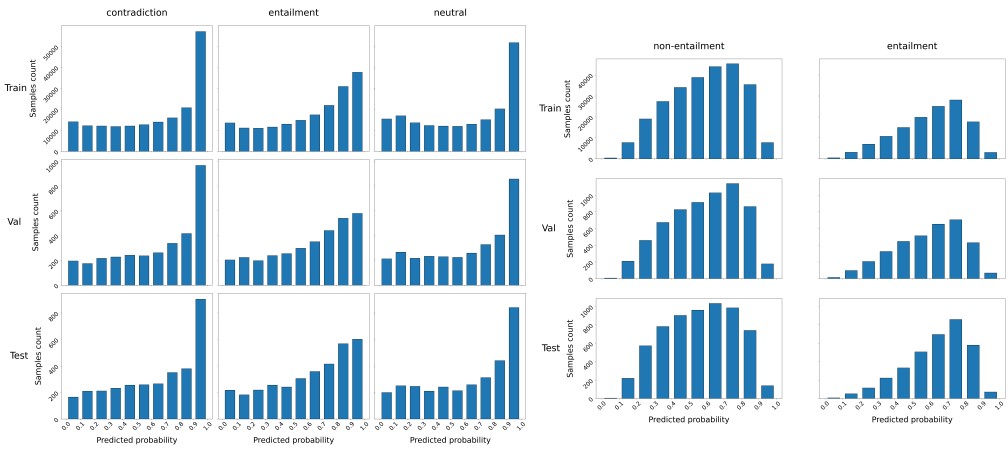

(a) Biased model for hypothesis bias. Train, val, and test are the SNLI train, dev, and test respectively.

(b) Biased model for overlap bias. Train, val, and test are the MNLI train, dev matched, and dev mismatched respectively.

Figure 5: Histogram of the probabilities the biased model assigned to the ground truth label. The rows indicate same set (train, validation, test) and the columns indicate the ground truth label of the samples.

In Section 3.2, we mentioned that two thresholds were used to generate the environments. $t_1$ was used as a threshold for the total variation distance of a score vector from the uniform distribution, and $t_2$ as a minimal required gap between the predicted probabilities assigned to the two most likely labels. These thresholds were qualitatively chosen and fixed throughout all the experiments, with $(t_1, t_2) = (0.205, 0.5)$ for hypothesis bias and $(t_1, t_2) = (0.11, 0.4)$ for overlap bias. Several considerations guided this choice. First, the thresholds were chosen so that the resulting environments are not too small. In each of the datasets, the accumulated size of the training subsets $\mathcal{D}_{id}$, $\mathcal{D}_{ood}$, and $\mathcal{D}_{un}$ used to generate the training environments was more than half the original size, with $\sim 380k$ for SNLI and $\sim 300k$ for MNLI. Next, since the performance of the biased model comes from the bias aligned samples, we tried to choose thresholds for which the proportion of bias aligned samples was close to the model's performance. Lastly, we had to have enough samples from each subset to be able to create the required environment probabilities, putting more restrictions on the choice of thresholds.

### D.3 Annotator agreement on bias misaligned subset

To understand the difficulty of each sample, we look at the 5 labels supplied for each of the samples in the SNLI test set. We refer to the number of labels matching the majority vote as the "majority count" and suggest that it can be indicative of the difficulty of the sample. That is to say, since for 5 labels in total the majority count can be 3, 4 or 5, we consider samples with a majority count of 3 or 4 to be more controversial, and therefore harder, than those with a majority count of 5. Indeed, in Table 10 we can see that the ratio of samples with a majority count of 3, i.e., which two annotators classified differently, in the bias misaligned subset is significantly larger than their proportion in the two other subsets.

Table 10: Using annotators agreement to represent noisiness of causal features in SNLI test subsets. Each row represents the distribution of samples in the subset across majority counts.

|                | 3    | 4    | 5    |
|----------------|------|------|------|
| unbiased       | 0.19 | 0.30 | 0.51 |
| bias aligned   | 0.11 | 0.28 | 0.60 |
| bias misaligned| 0.31 | 0.32 | 0.38 |

## D.4 Per class accuracy on the bias misaligned subset

Table 11a shows again the results on the MNLI test environments (which are in fact the bias aligned, bias misaligned and unbiased subsets) for overlap bias, discussed in Section 4.3. We look at per class accuracy on the bias misaligned subset in Table 11b. Recall that the non-entailed samples in this subset display overlap bias while the entailed samples display non-overlap bias. Noticing that performance on one subset decreases while the other increases, the performance gain seems to stem from mitigation of the bias correlated with the non-entailment class.

Table 11: Attributing IRM's performance gain on o.o.d data to mitigation of one of the biases.

(a) F1 macro on subsets of MNLI dev mismatched for overlap bias.

|     | Unbiased | Bias aligned | Bias misaligned subset |
|-----|----------|--------------|------------------------|
| ERM | 85.23 ($\pm$0.69) | 96.67 ($\pm$0.28) | 62.66 ($\pm$0.69) |
| IRM | 83.75 ($\pm$0.46) | 95.44 ($\pm$1.1) | 64.12 ($\pm$3.86) |

(b) Accuracy per class on the bias misaligned subset for overlap bias.

|     | non entailment | entailment |
|-----|----------------|------------|
| ERM | 80.73 ($\pm$1.83) | 48.97 ($\pm$5.87) |
| IRM | 77.84 ($\pm$3.0) | 59.66 ($\pm$15.77) |

## D.5 Results on o.o.d benchmarks

We evaluate models trained with ERM and IRM in the overlap bias setting on the HANS benchmark and report results in Table 12. Results show that IRM is outperformed by ERM on the not-entailed ($\neg E$) class in all heuristics. For the hypothesis bias setting we evaluate our models on the RP test set, on which the accuracy achieved is $58.58 \pm 0.97$ for ERM and $59.15 \pm 2.21$ for IRM, showing that ERM and IRM perform on par.

Table 12: Comparing accuracy per class on heuristics in HANS.

|     | Lexical overlap | | sub-sequence | | Constituent | |
|-----|-----------------|------|--------------|------|-------------|------|
|     | $\neg E$ | $E$ | $\neg E$ | $E$ | $\neg E$ | $E$ |
| ERM | 0.41 ($\pm$0.29) | 99.51 ($\pm$0.36) | 1.24 ($\pm$0.73) | 100.0 ($\pm$0.0) | 5.79 ($\pm$3.63) | 98.02 ($\pm$1.21) |
| IRM | 0.18 ($\pm$0.1) | 99.7 ($\pm$0.23) | 0.51 ($\pm$0.15) | 100.0 ($\pm$0.00) | 1.76 ($\pm$1.66) | 99.46 ($\pm$0.62) |