# OpenReview forum: "IRM—when it works and when it doesn't: A test case of natural language inference"
_NeurIPS.cc/2021/Conference — NeurIPS 2021 Poster_

### Official Review · Reviewer_Ktci · 2021-07-16

**Rating:** 6
**Confidence:** 3

**Summary:**

The paper presents an empirical study of IRM on NLI focusing on hypothesis and overlap bias. To do this, the paper defines the notion of bias "strength" p_e (probability of the bias label, given that the instance is indeed biased) and "prevalence" alpha_e (proportion of biased samples) for environment e. In practice this is controlled by training a biased predictor that only uses the (known) bias of the input to predict the label (e.g., for hypothesis bias we can finetune BERT on hypotheses only). Training environments correspond to different values of p_e and alpha_e (e.g., (0.7, 0.82) and (0.9, 0.82)) which is necessary for IRM training. The model is then tested on in-domain/bias aligned (e.g., alpha=1 and p=1), out-of-domain/bias misaligned (alpha=1 and p=0), and unbiased (alpha=0) data. The main finding of the paper is that: (1) IRM can outperform ERM even on natural data if the test set is bias misaligned, (2) the performance of IRM is heavily influenced by p, alpha, and also data size in training.

**Main Review:**

STRENGTHS

- As the paper says, this is one of the few studies on IRM that focus on non-synthetic data. Given the interest on IRM this is a useful resource.

- The paper's experimental design seems sound. NLI (binary formulation) is a good task to consider since it's simple and has known biases, though a flip side is it's sometimes viewed too synthetic to be a "real" task in NLP.

- That IRM can outperform ERM on natural data under a right setting is encouraging news.


WEAKNESSES

- Inevitably the scope is narrow. At this point we're trying to making something that doesn't work in most settings work mostly because it's theoretically justified and interesting. While the positive finding of the paper is good news, I'm not sure how much it matters outside the scope of IRM/maybe robust learning.

- This also raises some concern on practicality. Basically the finding (Section 5) seems to say that if bias is extremely prevalent in training data (alpha > 0.7) and extremely strong (p_e > 0.7-0.9), and we have plenty of data in each environment, then IRM can possibly improve a little over ERM, but even then the improvement is slight and training is unstable. The training domains are created by controlling bias prevalence and strength, and I'm not sure if it's feasible to satisfy these conditions in real-world data.


QUESTION

- The paper (and most existing works, I think) holds a particular type of bias fixed and uses different values of p_e and alpha_e to create IRM-friendly environments. Is this the only way to  use IRM, or is it possible to come up with a more natural setting where we don't assume the knowledge of bias (there may be unknown multiple types of bias) and still create training environments suitable for IRM?

**Time Spent Reviewing:**

2

---

> ### Author Response · Authors · 2021-08-10
> **Response to reviewer Ktci**
>
> We are happy to hear that the reviewer found the experimental setting satisfying and the results encouraging. We appreciate the reviewer’s comments and discussion and address the main concerns raised in the review.
>
> 1. “While the positive finding of the paper is good news, I'm not sure how much it matters outside the scope of IRM/maybe robust learning.”
>
> Please refer to the scope section in the general response.
>
> 2. “The training domains are created by controlling bias prevalence and strength, and I'm not sure if it's feasible to satisfy these conditions in real-world data.”
>
> The controlled experiments in section 5, where all three aspects of the environment are controlled (bias strength, prevalence and data size), are intended for analysis. The experiments presented in section 4, which target the performance of IRM in natural settings, only control bias strength. As long as we have a biased classifier (which is based on the known bias source), we can easily create such environments in real-world data.
>
> 3.  “Is this the only way to use IRM, or is it possible to come up with a more natural setting where we don't assume the knowledge of bias (there may be unknown multiple types of bias) and still create training environments suitable for IRM?”
>
> One approach is using weak learners, such as a low capacity model [1] (e.g., TinyBert [3]) or a model trained on a small subset of the data [2], to detect unknown dataset biases. These studies show promising results that could be extended to our case.
> Another approach is using different datasets or domains as environments. Although preliminary experiments with MNLI genres as environments did not show good results, this area could be further investigated.
>
> [1] Sanh, V., Wolf, T., Belinkov, Y. and Rush, A.M., 2020, September. Learning from others' mistakes: Avoiding dataset biases without modeling them. In International Conference on Learning Representations.
>
> [2] Utama, P.A., Moosavi, N.S. and Gurevych, I., 2020, November. Towards Debiasing NLU Models from Unknown Biases. In Proceedings of the 2020 Conference on Empirical Methods in Natural Language Processing (EMNLP) (pp. 7597-7610).
>
> [3] Turc, I., Chang, M.W., Lee, K. and Toutanova, K., 2019. Well-read students learn better: On the importance of pre-training compact models. arXiv preprint arXiv:1908.08962.

---

### Official Review · Reviewer_kZ64 · 2021-07-16

**Rating:** 7
**Confidence:** 4

**Summary:**

In this paper, the authors perform an empirical analysis of the effectiveness of Invariant Risk Minimization (IRM). They focus on the natural language inference (NLI) task, and while prior work examining the efficacy of IRM has focused on synthetic benchmarks, results from this paper support the hypothesis that models trained with IRM are generally less sensitive to spurious correlations even in naturalistic settings. The authors conduct a rigorous ablation study with various random seeds, dataset size, strength of spurious correlations, and the presence of spurious correlations and study the influence of each on the efficacy of IRM as applied to NLI. In general the paper presents a good analysis, however, certain generalizations made in the paper may not stand up to scrutiny, as I point below. I think this paper could be a good analysis paper but needs significant improvements to be accepted.

**Limitations And Societal Impact:**

I think the authors have done a decent job at scoping out the limitations and societal impacts. I think connecting your point about the assumption of having a "debiased" model in production and how it could be harmful without proper understanding or safety measures to prior work in fairness or philosophy could be helpful for the reader.

**Main Review:**

- The paper is an analysis paper so to me the novelty lies in the knowledge gained from the analysis. It is particularly exciting to see that IRM works as applied to naturalistic settings for NLI tasks. To the best of my knowledge, this might be the first paper to try and investigate the effectiveness of IRM on naturalistic datasets. It is great to see that the results support prior work that has looked at more synthetic settings.
- The problem setup is on the right track but needs to be fixed. Your findings are very interesting but you cannot show results on one (or two) out-of-domain evaluation set and claim that one model performs well out-of-domain compared to some other model. You could easily come up with another dataset where the other model performs better by just picking all the examples where the second model was correct and first was not, and that would be an out-of-domain dataset as well. Out-of-domain is not well defined, which is why it is necessary to show your results on a battery of out-of-domain datasets. Showing out-of-domain results on one dataset and claiming out-of-domain generalization is like showing correct prediction on one test example and claiming 100% accuracy iid. There are other ood datasets in NLI that you should also evaluate on, such as SNLI [1], ANLI [2], and counterfactually revised SNLI examples as collected in [3].
- Presentation of this paper needs significant improvements. It was not easy to follow your experimental design in particular (both Sections 3 and 4). Additionally, there are other issues such as in the analysis section, it would be nice if you support empirical findings with potential reasons as to why we might be observing something.

Additional comments:
- For your synthetic bias experiment, you’re appending certain tokens to various classes to inject synthetic bias. But since you’re using BERT, and it is a contextual model, it would be good to run these experiments with different tokens to weed out any effects a particular token might be having on the contextual representation at-large. Ideally you would expect the same results across all replications as you see right now, but we can’t be sure.
- I don’t think I quite follow Lines 222-225. Why would the test set have any role in whether a trained model relies on certain correlations or not?
- It is interesting that in your experiments, IRM showed a large variance in performance across random seeds but ERM did not. This is different from what was observed about ERM in prior work on stress test evaluations (see [4]). Do you have any thoughts on why this might be happening here?

[1] Bowman, Samuel R., Gabor Angeli, Christopher Potts, and Christopher D. Manning. "A large annotated corpus for learning natural language inference." In Conference on Empirical Methods in Natural Language Processing, EMNLP 2015, pp. 632-642. Association for Computational Linguistics (ACL), 2015.

[2] Nie, Yixin, Adina Williams, Emily Dinan, Mohit Bansal, Jason Weston, and Douwe Kiela. "Adversarial NLI: A New Benchmark for Natural Language Understanding." In Proceedings of the 58th Annual Meeting of the Association for Computational Linguistics, pp. 4885-4901. 2020.

[3] Kaushik, Divyansh, Eduard Hovy, and Zachary Lipton. "Learning The Difference That Makes A Difference With Counterfactually-Augmented Data." In International Conference on Learning Representations. 2020.

[4] Zhou, X., Nie, Y., Tan, H., & Bansal, M. (2020, November). The Curse of Performance Instability in Analysis Datasets: Consequences, Source, and Suggestions. In Proceedings of the 2020 Conference on Empirical Methods in Natural Language Processing (EMNLP) (pp. 8215-8228).


--------------------------------------------------------------------------------------
I thank the authors for the time they've taken to address my concerns. After careful deliberation and going back and forth on points raised by other reviewers, I am happy to increase my score to recommend acceptance. I would urge the authors to update their draft to address the concerns raised, including doing a deeper discussion of the negative results that are present in the Appendix.

**Time Spent Reviewing:**

9

---

> ### Author Response · Authors · 2021-08-10
> **Response to reviewer kZ64**
>
> We thank the reviewer for the in-depth review and are grateful for the insightful feedback. We address their specific concerns below, and we are happy to continue discussing any of these points or answer follow-up questions.
>
> 1. “you cannot show results on one (or two) out-of-domain evaluation set and claim that one model performs well out-of-domain compared to some other model...“
>
> While o.o.d is a very general concept that might refer to any distribution different from the training distribution, in this paper we focus on bias misaligned distributions, i.e., a distribution in which the correlation with the biased feature is flipped at train and test time. This is in agreement with the approach in prior work, for example [1,2] with colored MNIST experiments. It is unrealistic to expect a model to perform well on arbitrary distributions, therefore we restrict our analysis to bias misaligned distributions which capture if the model has unlearned the bias.
> We recognize that this might not have been clear enough from the introduction, and will clarify the definition of the o.o.d. distributions.
>
> [1] Arjovsky, M., Bottou, L., Gulrajani, I. and Lopez-Paz, D., 2020. Invariant Risk Minimization. stat, 1050, p.27.
> [2] Krueger, D., Caballero, E., Jacobsen, J.H., Zhang, A., Binas, J., Zhang, D., Le Priol, R. and Courville, A., 2021, July. Out-of-distribution generalization via risk extrapolation (rex). In International Conference on Machine Learning (pp. 5815-5826). PMLR.
>
> 2. “It was not easy to follow your experimental design in particular (both Sections 3 and 4).”
>
> We will improve the presentation of the experimental design in general. We also welcome any comments on specific parts that are unclear or hard to follow.
>
> 3. “...in the analysis section, it would be nice if you support empirical findings with potential reasons as to why we might be observing something.”
>
> Below we provide potential reasons to the empirical findings and will add them in the updated version.
> Bias prevalence - as more samples are either bias aligned or bias misaligned (i.e. contain a biased feature), we get more signal in the IRM penalty term indicating the instability of the biased feature across environments, making it easier for IRM to detect and discard it.
> Data size - As we mention in the paper (section 5.3), our results fall in line with the sample complexity analysis done in Ahuja et al. (2020), showing that IRM approaches the desired o.o.d solution in the finite sample regime as data size increases.
> Bias strength - while IRM seems to improve with increased prevalence and data size, it degrades as bias strength increases, but just not as much as ERM. This is in line with the discussion in lines 222-225, suggesting that while IRM is more robust to bias than ERM, it is not able to completely discard it, even in synthetic settings let alone natural, more complex settings. Therefore, as bias is stronger it is picked up more by both methods and only partially discarded by IRM.
>
> 4. “I don’t think I quite follow Lines 222-225. Why would the test set have any role in whether a trained model relies on certain correlations or not?”
>
> The test set has no role in whether a trained model relies on certain correlations. Evaluating the model on datasets that are increasingly misaligned (in terms of the correlation with the biased feature) demonstrates the reliance of both IRM and ERM on the bias to make their predictions.
>
> 5. “For your synthetic bias experiment, you’re appending certain tokens to various classes to inject synthetic bias. But since you’re using BERT, and it is a contextual model, it would be good to run these experiments with different tokens to weed out any effects a particular token might be having on the contextual representation at-large.”
>
> Our preliminary experiments show that other sets of bias tokens give similar performance as the one used in the paper. We will add such experiments in the revised version.
>
> 6. “It is interesting that in your experiments, IRM showed a large variance in performance across random seeds but ERM did not. This is different from what was observed about ERM in prior work on stress test evaluations (see [4]). Do you have any thoughts on why this might be happening here?”
>
> The results in [4] show that large variance is observed mainly on HANS and some of the stress test sets. These sets are extremely different from the training set, both with respect to the biased features but also with respect to other general language properties. This is in contrast to our o.o.d sets, which are o.o.d only with respect to the biased features, which may explain the relative stability of ERM. This is consistent with the stability [4] reported on SNLI hard, for example.
>
> 7. “I think connecting your point about the assumption of having a "debiased" model in production and how it could be harmful without proper understanding or safety measures to prior work in fairness or philosophy could be helpful for the reader.”
>
> We thank the reviewer for pointing out this connection, we will make sure to add it in the revised version.

---

### Official Review · Reviewer_64jW · 2021-07-17

**Rating:** 8
**Confidence:** 4

**Summary:**

The paper performs a case study of applying Invariant Risk Minimization (IRM) training to the task of recognizing Natural Language Inference (NLI). The goal is to test whether IRM can be helpful for training bias-free models on biased data. The considered biases include the hypothesis bias, the word overlap bias and a special synthetic bias whereby a noisy gold label is directly appended to the input. The training environments are constructed to feature different high levels of bias strength, i.e. the degree to which the biased feature is predictive of the label B(x). The test environments are constructed in such a way that the biased feature is a bad predictor of the label. The experiments show that IRM brings a significant but small improvement on top of ERM. The paper also features extensive analysis on the impact of bias strength, bias prevalence and training data size on the performance difference between IRM and ERM.


**Limitations And Societal Impact:**

As the paper studies methods to reduce bias, I can’t see negative societal consequences, only positive ones.


**Main Review:**

Significance: The paper studies a very important research question: how to train bias free models. Perhaps the most interesting result is the one reported in Table 3: IRM training does not help to fully ignore even a very clear and carefully controlled synthetic bias.

Clarity: The paper is largely clear. Improvements in the following aspects could help:
- It would be great to have a table featuring $\alpha_e$, $p_e$ and training data size for all experiments.
- It is not 100% clear what ERM models are trained on. Mixture of training data from the two environments?
- It is not very clear what exactly is the classifier and what is the representation function in the NLI experiments.

Quality: the paper is very well executed. The study appears complete and the conclusions are clearly formulated.

Originality: I do not know IRM literature well enough to confidently say if experiments with pretrained language models and NLP tasks have never been done before. I’m not aware of such studies, hence I find the paper conclusions novel and original.

As the paper studies methods to reduce bias, I can’t see negative societal consequences, only positive ones.


**Time Spent Reviewing:**

3

---

> ### Author Response · Authors · 2021-08-10
> **Response to reviewer 64jW**
>
> We are happy to hear that the reviewer found the paper clear, complete, and the conclusions novel and original. We appreciate the reviewer’s comments and discussion and address the main concerns raised in the review.
>
> 1. “It would be great to have a table featuring αe, pe and training data size for all experiments.”
>
> Thank you for the comment, we will add such a table in the revised version.
>
> 2. "It is not 100% clear what ERM models are trained on. Mixture of training data from the two environments?"
>
> Yes, the ERM models are trained on a mixture of training data from the same two environments. We will make this point more clear in the paper.
>
> 3. "It is not very clear what exactly is the classifier and what is the representation function in the NLI experiments."
>
> Following the re-formulation of IRM as the practical IRMv1 in the original paper, we are using a dummy classifier (w=1.0 when the task is binary classification and extend to a vector of ones for multi class classification) on top of logits generated by a fine-tuned BERT model. For the toy setting we use a linear layer on top of the mean of character embeddings to produce the logits. We will make sure to make this distinction more clear in the revised version.

---

> > ### Comment · Reviewer_64jW · 2021-08-11
> > **thanks**
> >
> > Thank you for your response!

---

### Official Review · Reviewer_qz54 · 2021-07-17

**Rating:** 5
**Confidence:** 4

**Summary:**

This paper designs natural language inference (NLI) experiments to investigate the properties of the bias mitigation method, Invariant Risk Minimization (IRM). The authors design three out-of-domain setups: synthetic data and synthetic bias, natural data and synthetic bias, and natural data with real bias, to provide detailed analysis on IRM and ERM, under different conditions such as changing bias strength, prevalence, data sizes, and random seeds. The paper advocates the need for more naturalistic settings to study the state-of-the-art bias mitigation models.

**Ethical Concerns:**

No concerns

**Main Review:**

The paper highlights the need for more naturalistic settings to study a state-of-the-art bias mitigation model IRM, and makes a concrete step towards that. The work is well motivated and is well set up in a task (and datasets) where spurious correlation is known to be an issue. The paper carefully constructs the experiments, and performs detailed experiments and analyses that well support the conclusions made on the NLI datasets, contributing to the empirical understanding of IRM in a specific naturalistic setting. The method proposed to construct naturalistic biased datasets is interesting. In general, the paper is clear and easy to follow.

The paper did not propose new models. The originality in that respect is very limited. As an empirical study (on an existing model), I would like to see experiments on more tasks (instead of just NLI) which would help make the conclusions more convincing. In my opinion, this seriously limits the significance of the study.

I am not fully convinced that the experiment setup represents the synthetic-to-naturalistic continuum for NLI, since the synthetic-data-and-synthetic-bias toy set has little property of “NATURE LANGUAGE” inference, and I am not convinced it is “a simplified NLI task”, e.g., considering the properties of SNLI and MNLI datasets (e.g., these datasets focus a lot on lexical semantic relations, e.g., hypernyms, and has little XOR property.)

I am also concerned with whether the methods used to construct the environments (i.e., environments generation) can be easily adapted to study more complicated bias where manually designing features is hard. I am also concerned if they can be scaled up to consider more diverse biases (e.g., sentence compositional structures can also be the source of bias) and to consider interaction of multiple sources of biases.

Can the author(s) provide some discussion on this research with regard to the recent work (Kamath et al., 2021)?


**Time Spent Reviewing:**

8

---

> ### Author Response · Authors · 2021-08-10
> **Response to reviewer qz54**
>
> We are pleased to hear the reviewer found the paper clear, well motivated and designed, and the method proposed to construct naturalistic biased datasets interesting. We appreciate the reviewer’s comments and discussion and address the main concerns raised in the review below.
>
> 1. “The paper did not propose new models. The originality in that respect is very limited. As an empirical study (on an existing model), I would like to see experiments on more tasks (instead of just NLI) which would help make the conclusions more convincing.”
>
> Please refer to the scope section in the general response.
>
> 2. “I am not fully convinced that the experiment setup represents the synthetic-to-naturalistic continuum for NLI, since the synthetic-data-and-synthetic-bias toy set has little property of “NATURE LANGUAGE” inference, and I am not convinced it is “a simplified NLI task” “
>
> While toy settings are far from representing the complexity of a real task such as NLI, they have been widely used to enable careful analysis and full control of experiments. Much work in NLP uses toy settings of similar spirit to simulate natural language. For example, [1,2] use integer sequences with weak and strong features (presence of an integer vs more complicated pattern like first and last number are the same) to design a binary sequence classification task.
> Many use cases in natural language require some sort of pattern matching, similar to the one in our toy setting. For example, the sentence “a cat is sleeping” suggests that there is “a cat”, which requires such pattern matching. The specific toy setting we apply was not only used in Belinkov et al. 2019b, but also inspired other toy experiments such as the one presented in [3].
>
> [1] Jha, R., Lovering, C. and Pavlick, E., 2020. Does Data Augmentation Improve Generalization in NLP?. arXiv preprint arXiv:2004.15012.
> [2] Lovering, C., Jha, R., Linzen, T. and Pavlick, E., 2020, September. Predicting Inductive Biases of Pre-Trained Models. In International Conference on Learning Representations.
> [3] Ravichander, A., Belinkov, Y. and Hovy, E., 2020. Probing the probing paradigm: Does probing accuracy entail task relevance?. arXiv preprint arXiv:2005.00719.
>
> 3. “I am also concerned with whether the methods used to construct the environments (i.e., environments generation) can be easily adapted to study more complicated bias where manually designing features is hard. I am also concerned if they can be scaled up to consider more diverse biases”
>
> Although for overlap bias we used manually designed features, the features for the hypothesis bias were learned. In general, as long as the bias is known we can learn a biased classifier, even for complex high-dimensional biases or multiple bias types. In the case of unknown biases, we can use weak learners as bias models, which has been shown to be effective for debiasing models and to discover known dataset biases such as hypothesis bias ([1,2]).
>
> [1] Sanh, V., Wolf, T., Belinkov, Y. and Rush, A.M., 2020, September. Learning from others' mistakes: Avoiding dataset biases without modeling them. In International Conference on Learning Representations.
>
> [2] Utama, P.A., Moosavi, N.S. and Gurevych, I., 2020, November. Towards Debiasing NLU Models from Unknown Biases. In Proceedings of the 2020 Conference on Empirical Methods in Natural Language Processing (EMNLP) (pp. 7597-7610).
>
> 4. Can the author(s) provide some discussion on this research with regard to the recent work (Kamath et al., 2021)?
>
> Kamath et al. (2021) use a simple toy example to point out the discrepancies between the original IRM formulation and various instantiations of it. Since in non-toy examples such comparisons between different methods is non-trivial, we focus on IRMv1, the practical formulation suggested in the original paper.
> Kamath et al. also discussed sample complexity for the different formulations. Our results in section 5 are consistent with their finding that IRMv1 improves with increasing data size.

---

### Author Response · Authors · 2021-08-10
**General response**

We thank all the reviewers for their helpful comments and insights.
Below we address two common issues raised by the reviewers.

1. The scope of this work is limited.

We focus on overcoming spurious correlations (bias) in the data, which is a key challenge towards trustworthy ML and has been widely studied in both the NLP and ML communities.

Why IRM? IRM is an important framework in robust ML and has spurred a line of follow-up works because of its causal inference foundation and relatively fewer assumptions. However, almost all prior work has focused on synthetic settings or theoretical analysis. As pointed out by reviewers Ktci, kz64, and 64jw, we are among the first to investigate the effectiveness of IRM in natural settings (realistic biases).

Why NLI?  NLI is a widely recognized task in natural language understanding. In addition, it’s one of the most studied tasks for model robustness; the prominent and diverse biases make controlled experiments easier. We argue that the results are not restricted to NLI, because the biases we studied for NLI datasets are representative and also found in other tasks. For example, hypothesis bias has been found in VQA where models use only the question to answer, disregarding the image [1]; and in fact verification where it is known as claim-only bias [2]. Similarly, overlap bias has been found in paraphrase identification [3].

Depth vs Width. Instead of experimenting with more tasks (with similar biases), we chose to dive deep into NLI and investigate various aspects of biases including synthetic vs natural, bias types, prevalence, strength, and dataset sizes, as noted by reviewer kz64.

[1] Agrawal, A., Batra, D., Parikh, D. and Kembhavi, A., 2018. Don't just assume; look and answer: Overcoming priors for visual question answering. In Proceedings of the IEEE Conference on Computer Vision and Pattern Recognition (pp. 4971-4980).

[2] Schuster, T., Shah, D., Yeo, Y.J.S., Ortiz, D.R.F., Santus, E. and Barzilay, R., 2019, November. Towards Debiasing Fact Verification Models. In Proceedings of the 2019 Conference on Empirical Methods in Natural Language Processing and the 9th International Joint Conference on Natural Language Processing (EMNLP-IJCNLP) (pp. 3419-3425).

[3] Zhang, Y., Baldridge, J. and He, L., 2019, June. PAWS: Paraphrase Adversaries from Word Scrambling. In Proceedings of the 2019 Conference of the North American Chapter of the Association for Computational Linguistics: Human Language Technologies, Volume 1 (Long and Short Papers) (pp. 1298-1308).


2. Significance of the study

On IRM/robust learning: Our results complement both existing theoretical and empirical studies on IRM. Specifically, we formalize the concepts of bias prevalence and bias strength, which are shown to be key factors influencing the performance of IRM. Bias prevalence was overlooked in prior theoretical analysis and the effect of bias strength has not been empirically investigated. We believe these concepts will guide future work on IRM.

New method: Prior work in IRM splits environments using categorical biased features. However, in NLP problems, the biased feature is usually high-dimensional (e.g., the hypothesis). We have developed a simple way to split data into different environments targeting known biases (acknowledged by reviewer qz54), which is applicable to any setting (especially the real-world settings like what we studied) where the bias source is known (but the bias feature may be complex variables).

Influence on the community: We think that an important takeaway of this research is to demonstrate the importance of more natural and flexible characterization of experiments, both in theoretical and empirical research; and to call for more focus on natural settings in robust learning and o.o.d. generalization.

---

> ### Author Response · Authors · 2021-08-30
> **General response 2**
>
> We are happy to answer any additional questions regarding the work.
> Please let us know if the response addresses your concerns.

---

### Decision · Program_Chairs · 2021-09-27

**Decision:**

Accept (Poster)

**Comment:**

This paper studies the effectiveness of Invariant Risk Minimization (IRM) in order to train unbiased classifiers in a specific Natural Language Inference (NLI) setting. The authors devise three experimental scenarios: synthetic data and synthetic bias, natural data and synthetic bias, natural data and natural bias. In all three settings, the authors study whether IRM is capable of training bias free classifiers by controlling the prevalence, strength of bias and data size. Results seem to show that, apart from a very synthetic setting, IRM seems to only marginally succeed in training robust classifiers, especially when bias prevalence, strength and data sizes are small.

—

All reviewers concur that this paper has extensive experiments and can be a useful resource as is one of the few studies of IRM with non-synthetic data. The paper appears clear although some sentences must be reformulated and made more precise/formal (e.g. "IRM is not able to completely ignore the bias when the test environments are sufficiently different"). Experiments are well-executed. One of the concerns of the reviewers was that the paper is somewhat limited in scope: binary NLI may be a too simplistic task in NLP and thus it is unclear whether the results in the paper can tell something about IRM’s performance in other settings. A reviewer also noticed the absence of performance in more ‘standard’ out-of-domain test sets (e.g. ANLI, counterfactual SNLI, …). We concurred that, in these test sets, it might be difficult to control for factors influencing IRM performance, and it might be reasonable to limit this paper to the careful study of controlled settings. After discussion with reviewers, I suggest the authors to move the HANS results from Appendix C.5 into the main text as this is a well-known evaluation set (even if results might not be directly comparable with reported baselines) and address some of the aforementioned issues. The authors might consider to also test on the Revised Premise dataset from [1]. Finally and most importantly, one source of confusion revolved around the takeaways of the paper. Reviewers and myself agree that this seems like a carefully executed negative results paper, as IRM doesn't seem very successful apart from the synthetic setting. I strongly suggest the authors to take a clearer stance on this by removing emphasis (i.e. from the abstract) on the fact that IRM performs well in a synthetic setting (which is mildly interesting) and emphasize the message in lines 50-51: "However, in these more naturalistic settings, IRM is not able to completely discard the bias, while ERM does not rely solely on the bias. Thus, in practice the advantage of IRM is small."

Overall, this paper is a useful resource as it's one of the few that studies IRM in a controlled and natural, albeit simplistic, task. Provided that the authors incorporate in the final version the suggestions above and reviewers' feedback, I recommend this paper for acceptance.

[1] Kaushik, Divyansh, Eduard Hovy, and Zachary Lipton. "Learning The Difference That Makes A Difference With Counterfactually-Augmented Data." In International Conference on Learning Representations. 2020.